



# Exploring biases in brown carbon model representation with in-situ flight observations

Maegan A. DeLessio[1,2], Kostas Tsigaridis[3,2], Susanne E. Bauer[2,3]

[1]Department of Earth and Environmental Sciences, Columbia University, New York, NY 10025, USA
[2]NASA Goddard Institute for Space Studies, New York, NY 10025, USA
[3]Center for Climate Systems Research, Columbia University, New York, NY 10025, USA

*Correspondence to*: Kostas Tsigaridis (kostas.tsigaridis@columbia.edu)

**Abstract.** The inclusion of brown carbon (BrC), the subset of organic aerosols (OAs) that absorb light in the near UV to visible wavelength range, in climate models is necessary to represent an abundant atmospheric constituent and to improve
OA radiative forcing estimates. However, the large degree in variability of laboratory and field measurements of BrC properties makes model representation difficult. We utilized in-situ observations of BrC absorption from the DC3, SEAC⁴RS, ATom, WE-CAN, and FIREX-AQ flight campaigns to evaluate the GISS ModelE Earth system model BrC scheme. We focused on measurements influenced by biomass burning (BB) and the region of temperate and boreal North America. Average vertical profile comparisons of measured versus simulated absorption, corrected for model biases in
carbon monoxide and black carbon (commonly co-emitted with BrC) concentrations, revealed a systematic underestimation in modelled BrC absorption. To explore possible causes of this bias, we evaluated the model's BrC-to-OA relationship. Sensitivity tests were run to determine if parameter changes could improve model performance and therefore substantiate the potential causes of bias identified. Increasing model organic aerosol-to-organic carbon (OA-to-OC) mass ratio for BB and aged OAs greatly improved alignment to measured OA-to-OC but decreased OA burden and increased BrC bias in the upper
troposphere. Decreasing wet removal of BrC appeared to partially address the bias in aged air masses and fire plumes, while other potential BrC-specific biases, like missing secondary sources, could not be substantiated. Based on this, we highlight BrC processes that require further research and future directions for model development.

## 1 Introduction

Carbonaceous aerosols consist of black carbon (BC) and organic aerosols (OAs). BC is emitted from combustion of fuel and
biomass, while OAs originate from both biomass burning (BB) and secondary organic aerosol (SOA) formation via volatile organic compound (VOC) partitioning (Ito and Penner, 2005; Shrivastava et al., 2017; Mahilang et al., 2021). Though they have similar sources, the two aerosol species have distinct effects as short-lived climate forcers: BC is strongly absorbing (Jacobson, 2001; Ramanathan and Carmichael, 2008; Bond et al., 2013), with an estimated effective radiative forcing (ERF)



of 0.11 W m$^{-2}$, while OAs cool the atmosphere with an estimated ERF of -0.21 W m$^{-2}$ (Szopa et al., 2021). OAs are expected

to grow in importance, as warming temperatures and changes in precipitation drive increased wildfires (Flannigan et al., 2009; Keywood et al., 2013), SOA burdens grow (Tsigaridis and Kanakidou, 2007), and emission controls lead to reductions in anthropogenic aerosols sources (Bauer et al., 2022). Despite this, OAs pose a large gap in aerosol modeling, with an estimated ERF uncertainty of ±0.23 W m$^{-2}$, greater than the magnitude of the forcing itself (Szopa et al., 2021). To improve estimates of OA cooling, the subset of OAs that absorbs light, brown carbon (BrC), must be represented in climate models.

BrC is a classification of aerosols, rather than a specific compound or compounds class. It is characterized by its spectrally-dependent absorption in the near-UV to visible wavelength range, with less absorption at longer visible wavelengths (Laskin et al., 2015). Like other OAs, it is emitted by smoldering fires and other incomplete combustion (McMeeking et al., 2009; Chakrabarty et al., 2010), but can also form as an SOA from gaseous or aqueous precursors (Lee et al., 2014). Observations of BrC have shown its properties have large spatial and temporal variability. For instance, the ratio

of BrC to non-absorbing OA in BB emission varies across different vegetation biomes (Jo et al., 2016), and its imaginary refractive index (RI) varies with combustion conditions and fuel type, in the case of primary BrC (Fleming et al., 2020). BrC also undergoes aging in the atmosphere, with changes in absorption, volatility, and molecular weight over time (Di Lorenzo and Young, 2016; Di Lorenzo et al., 2017; Hems et al., 2021). In terms of absorption, BrC has been shown to brown, leading to increased absorption (Cheng et al., 2020; G. Schnitzler et al., 2020; Hems et al., 2020), as well as bleach, leading to

decreased absorption (Zhao et al., 2015).

Such variability in observed properties makes model representation of BrC difficult. To simulate BrC, models make assumptions and use simplified parameterizations to represent aerosol properties and aging. This, combined with variable baseline model configurations, results in a large range of estimated BrC radiative effect: 0.03-0.57 W m$^{-2}$ (Feng et al., 2013; Lin et al., 2014; Wang et al., 2014; Saleh et al., 2015; Hammer et al., 2016; Jo et al., 2016; Brown et al., 2018; Wang et al.,

2018; Tuccella et al., 2020; Zhang et al., 2020; Carter et al., 2021; Drugé et al., 2022; Xu et al., 2024). Continued evaluation of models with observations and retrievals can help constrain BrC schemes and, therein, BrC radiative effect estimates. The majority of BrC modelling studies use data from the ground-based Aerosol Robotic Network (AERONET; Sinyuk et al., 2020) to evaluate scheme performance (Feng et al., 2013; Lin et al., 2014; Wang et al., 2014; Jo et al., 2016; Brown et al., 2018; Wang et al., 2018; Tuccella et al., 2020; Zhu et al., 2021; Drugé et al., 2022; Skyllakou et al., 2024; Wang et al., 2024;

Xu et al., 2024).

In a previous study, we introduced BrC into the NASA GISS ModelE Earth system model (ESM; referred to as ModelE) (DeLessio et al., 2024a), then evaluated total model performance with our scheme against AERONET, as well as Moderate Resolution Imaging Spectroradiometer data (MODIS; Levy et al., 2013),. We found we were unable to constrain the BrC scheme with these data, primarily due to a limitation in ModelE's radiation scheme: ModelE computes radiative properties,

including aerosol optical depth (AOD) and absorbing aerosol optical depth (AAOD) over broad radiation bands, rather than distinct wavelengths (Bauer et al., 2010). For the UV-to-visible (UV-VIS) wavelength range–the BrC relevant range–



ModelE produces radiative properties over the entire visible band. These properties are indicative of 550 nm, the spectrally-weighted average of light from 300-770 nm. Our results showed that a BrC scheme did not have a discernible effect on ModelE total AOD and AAOD at this wavelength, inhibiting our ability to constrain the scheme's properties. We were not

able to use the absorption Ångström exponent (AAE) retrieval product from AERONET, relevant due to the spectrally-dependent absorption pattern of BrC, because that cannot be computed in ModelE within a radiation band.

Following this work, we used a retrieval of BrC from AERONET to evaluate the ModelE scheme (DeLessio et al., 2024b). In this subsequent study, we were able to constrain our scheme by aligning ModelE's BrC physical and optical properties as closely as possible to those assumed by the retrieval.  However, this speciated AERONET retrieval used

simplified parametrizations to represent BrC (Schuster et al., 2016), meaning it was not indicative of in-situ BrC chemistry and microphysics. For a more "physically correct" evaluation, we need to look to in-situ measurements of BrC. Laboratory studies have measured in-situ BrC properties, but they utilize either individual compounds as proxies or singular fuel sources in lab-burns (Saleh et al., 2014; Di Lorenzo and Young, 2016; Liu et al., 2016; Tang et al., 2016; Lin et al., 2018; Al Nimer et al., 2019; Shetty et al., 2019; Wong et al., 2019; Li et al., 2020). For this work, we focus on flight campaigns that can

measure BrC as it is emitted and evolves in the atmosphere, for a more in-depth in model scheme evaluation.

This work follows several studies that have either evaluated their BrC schemes against in-situ measurements from flight campaigns (Wang et al., 2018; Zhang et al., 2020; Carter et al., 2021), or have used them to estimate BrC effect through radiative transfer calculations (Zeng et al., 2020). Past analyses of such measurements have shown an enhancement of BrC absorption, relative to that of BC, in the upper troposphere (Zhang et al., 2017). While the precise mechanism behind this

enhancement is not known, some studies propose it's a result of distinct BrC behavior in deep convection (e.g. different processing and removal within a convective cloud environment) or in-cloud production of brown SOA (Wu et al., 2024; Zhang et al., 2020). As observations suggest BrC may be an important climate forcer relative to BC at higher altitudes, it is necessary for models to capture not just total column properties, but also the vertical distribution of BrC. This further supports the use of flight campaigns, which take measurements at varying altitudes, for model evaluation, as they can

provide information unavailable in the column products of AERONET and MODIS.

In this study, we present a BB-focused evaluation of ModelE's BrC scheme with in-situ BrC data from five flight campaigns: the Deep Convective Clouds and Chemistry (DC3) campaign, the Studies of Emissions, Atmospheric Composition, Clouds and Climate Coupling by Regional Surveys (SEAC4RS) campaign, the Atmospheric Tomography Mission (ATom), the Western Wildfire Experiment for Cloud Chemistry, Aerosol, Absorption and Nitrogen (WE-CAN), and

the Fire Influence on Regional to Global Environmental Experiment-Air Quality (FIREX-AQ). Since each campaign measured other aerosol species simultaneously with BrC, our analysis focuses not just on the vertical profiles of BrC aerosols, but also on the relationships between BrC, total OAs and BC. We first examined model simulated BC compared to campaign measurements to identify general biases in BB aerosols. Then, correcting for model BC bias, we evaluated the base case of BrC representation. Finally, we varied BrC and total OA properties in a series of sensitivity tests. The resulting



analyses identify possible sources of bias in ModelE's BrC and OA schemes and propose improvements in both to reduce them.

## 2 Methods

### 2.1 Aircraft measurements

We discuss here, in chronological order, each flight campaign from which aircraft measurements were used for model
evaluation. The Deep Convective Clouds and Chemistry (DC3) campaign was conducted over the central and south-eastern U.S. from May to June of 2012 (Barth et al., 2015). The focus of the campaign was storms, with sampling targeting convective inflow and outflows of storms. To evaluate ModelE's BrC scheme, we made use of measurements made on the NASA DC-8 aircraft of OA absorption (which we treat as BrC) and water-soluble organic carbon (WSOC) mass. BrC absorption was measured from liquid extracts of Teflon filters, sampled every 5 minutes with a 4.1 μm aerodynamic
diameter cut-off. For offline analysis, filters were first extracted in water then methanol. Absorption of these extracts were measured in the 200 to 800 nm range, and the water-soluble (WS) and methanol-soluble (MS) absorption at 365 nm was reported in units of $Mm^{-1}$. WSOC mass concentration was measured from the water extracts in a liquid waveguide capillary cell coupled to a total organic carbon analysis (LWCC-TOC) (Zhang et al., 2017).

To assess whether ModelE captures the relationships between BrC and co-emitted species, we also used total OA mass
concentration, OA-to-OC ratio, BC mass concentration, and carbon monoxide (CO) concentration measurements. OA mass and OA-to-OC ratio were measured online with a high-resolution time of flight Aerodyne aerosol mass spectrometer (HR-ToF-AMS; DeCarlo et al., 2006). Black carbon mass was measured online with a single-particle soot photometer (SP2; Schwarz et al., 2006) and CO was measured online via diode laser spectrometry (Sachse et al., 1987). Finally, to help narrow our analysis to samples impacted by BB, we used the PALMS BB aerosol number fraction, a variable determined through
particle analysis by laser mass spectrometry, where single aerosol particles are classified into several particle type (Thomson et al., 2000). The Studies of Emissions, Atmospheric Composition, Clouds and Climate Coupling by Regional Surveys (SEAC[4]RS) campaign was conducted over a similar region as DC3, from August to September of 2013 (Toon et al., 2016). The goal of this campaign was to investigate atmospheric composition over North America. As such, it included NASA DC8 flights that studied smoke from wildfires in western forests and from agricultural fires in the Mississippi Valley. We used the
same measurements, each obtained with similar techniques, from SEAC[4]RS as DC3 (Zhang et al., 2017).

The Atmospheric Tomography Mission (ATom) conducted flights along the central Pacific and Atlantic oceans, systematically profiling from the near surface up to approximately 13 km above sea-level (Thompson et al., 2022). There was a total of four ATom deployments, with BrC absorption measurements made in ATom-2 (January to February 2017), ATom-3 (September to October 2017), and ATom-4 (April to May 2018). BrC and WSOC ATom measurement techniques
were generally consistent with those of DC3 and SEAC[4]RS. Filter samples were taken at <5-minute intervals at lower altitudes and at a maximum interval of 15 minutes at higher altitudes (Zeng et al., 2020). Unlike earlier campaigns,



absorption was reported at multiple wavelengths between 200 and 800 nm, allowing for a calculation of absorption Ångström exponent (AAE). Only WS BrC absorption was used for evaluation, consistent with Zeng et al. (2020) who noted high blanks associated with methanol extraction. While ATom employed the same measurement techniques for OA mass, 130   BC mass, and BB number fraction as previous campaigns, CO was measured with a Picarro cavity ring-down spectrometer (McKain and Sweeney, 2021). ATom-2 had limited spatial coverage of BrC absorption above the limit of detection (LOD), primarily covering the mid-Atlantic, and did not include WSOC measurements. As such, it was removed from consideration in our study; since ATom-3 and ATom-4 took samples over the mid-Atlantic region, the main region covered in ATom-2 can still be studied using later deployments.

The Western Wildfire Experiment for Cloud Chemistry, Aerosol, Absorption and Nitrogen (WE-CAN) investigated the emissions and evolution of gases and aerosols from wildfire sand prescribed burns, primarily over the western U.S., from July to August of 2018 (Sullivan et al., 2022). This campaign included online measurements of BrC absorption and WSOC mass taken during C-130 research flights; this is the only campaign in which offline filter measurements were not collected. Instead, a particle into liquid sampler was used to collect aerosols (Zeng et al., 2021). The liquid sample was then passed 140   through an LWCC-TOC for absorption and WSOC measurement. Because this used a particle into liquid sampler, rather than filter extracts, BrC measurements were limited to WS absorption only. OA measurements were made with an AMS (Garofalo et al., 2019), BC was measured with SP2 (specifically using a counter-flow virtual impactor; Shingler et al., 2012), and CO was measured with a quantum-cascade laser (Ren et al., 2012). BB number fraction was not reported for WE-CAN, but such measurements aren't necessary for our analysis, as the campaign design inherently produced samples influenced by 145   BB.

The final campaign we used for evaluation was the Fire Influence on Regional to Global Environmental Experiment-Air Quality (FIREX-AQ) campaign (Warneke et al., 2023). The purpose of this comprehensive campaign was to obtain detailed measurements of trace gas and aerosol emission from wildfires and prescribed burns using aircraft, satellite, and ground-based measurements. FIREX-AQ included NASA DC8 flights designed to characterize fire plumes, deployed over the 150   western and south-eastern U.S. This campaign took both online and offline measurements of BrC, but to be consistent with other campaigns and allow for both WS and MS absorption data (unlike WE-CAN), we are using DC8 filter samples (Zeng et al., 2022). These were subject to the same offline analysis as DC3, SEAC[4]RS, and ATom. Like the latter, BrC absorption is provided at multiple wavelengths, allowing for a calculation of AAE (Zeng et al., 2021). Measurement of co-emitted species was consistent with DC3 and SEAC[4]RS campaigns, with OA measured via AMS, BC via SP2, and CO via diode 155   laser spectrometry. Unlike these campaigns, however, OC-to-OA ratio was not available in FIREX-AQ merged data. Like WE-CAN, BB number fraction was not reported and not needed, as this campaign focused on BB plumes. The list of variables we used for evaluation from each campaign, along with their associated measurement uncertainties (when available) are provided in Table 1.



**Table 1.** Reported (either in published papers about flight campaigns or in information about flight instruments) uncertainties of measured
variables used in this study. N/A is specified if the data wasn't collected or used for analysis in this study. We were unable to find
uncertainty information for several WE-CAN variables, so these boxes have been left blank (black).

| Campaign | Water soluble BrC absorption | Methanol soluble BrC absorption | Water soluble organic carbon (WSOC) conc. | Organic aerosol (OA) conc. | Black carbon (BC) conc. | Carbon monoxide (CO) conc. | BB number fraction |
|---|---|---|---|---|---|---|---|
| DC3 | 15% | 15% | 8% | | 30% | 2% | 0.15% |
| SEAC[4]RS | 18% | 20% | 8.8% | 38% | | 5% | |
| ATom-3 | 15.2% | 13.6% | 16.7% | | 25% | 3-5 ppb | N/A |
| ATom-4 | 17.4% | 20.6% | 15.9% | | | | |
| WE-CAN | 12% | N/A | ███████ | ███████ | ███████ | ███████ | ███████ |
| FIREX-AQ | 16% | 19% | 17% | 38% | 20% | 2% | N/A |

Some additional treatment of BrC absorption measurements was required for comparison to model output, as the measurements we use were absorption in dissolved solution ("bulk"), not ambient suspended particle absorption. To estimate ambient BrC absorption, we follow the procedure used by previous studies and multiply bulk measurements by a factor of 2 (Zhang et al., 2017; Wang et al., 2018; Zeng et al., 2020). Additionally, when MS BrC measurements aren't available (for instance in ATom and WE-CAN campaigns) or are below instrument limit-of-detection, we follow previous studies, assume the WS fraction makes up about 50% of total absorption, and thus multiply that absorption by 2. So, in the case where only WS BrC bulk absorption is reported, we would multiply that value by a factor of 4 to get from WS to total, and then from bulk to ambient absorption. If WS BrC is also not available, we exclude the entire datapoint. The factor of 2 for WS to total absorption conversion is inherently uncertain, as studies have reported this ratio ranging from 25% to 80% (Zeng et al., 2020), and some have specifically noted that water-insoluble (WI) BrC is more absorbing than WS (Laskin et al., 2015; Lin et al., 2018). If that latter is true, then a factor of 2 may bias the total measured BrC absorption low. The magnitude and spatial distribution of resulting BrC absorption data from each campaign, filtered for values below LOD and converted to total ambient absorption, can be seen in Figs. 1 and 2.



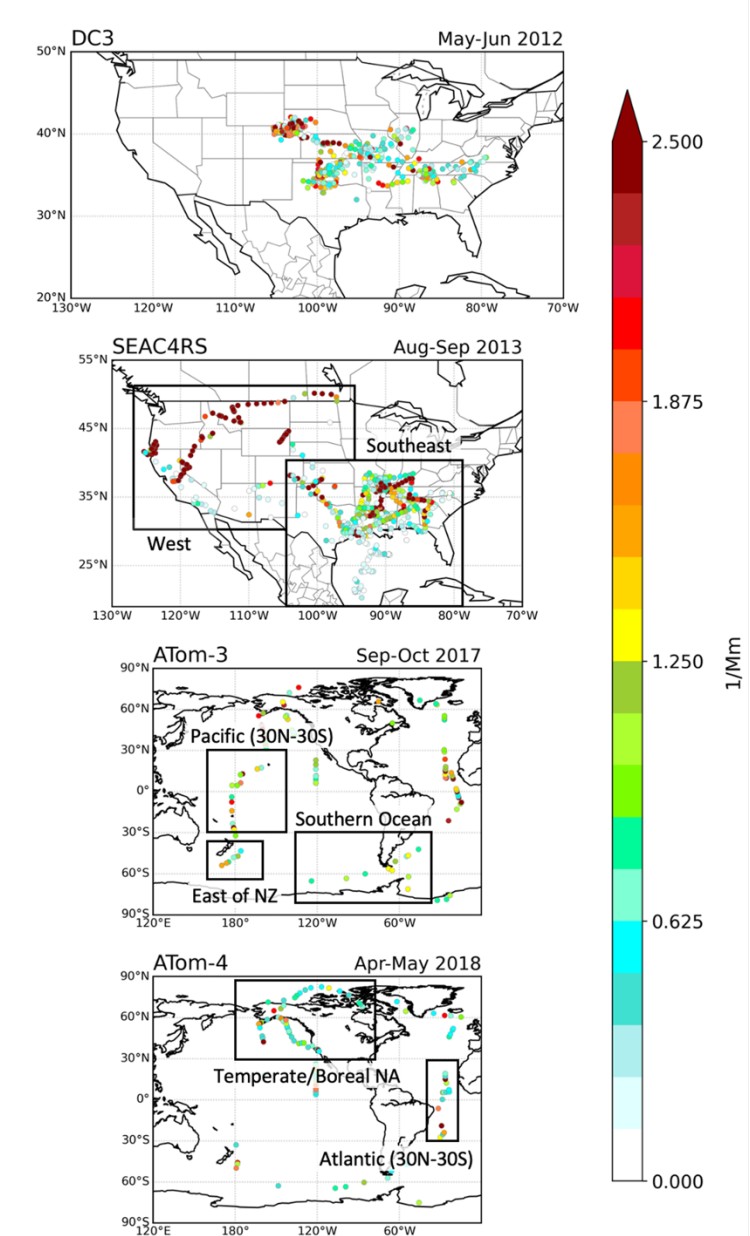

**Figure 1.** Map of BrC absorption coefficient measurements, at 365 nm, collected during the DC3, SEAC[4]RS, ATom-3, ATom-4 flight campaigns. Total ambient absorption is calculated from water soluble (WS) and, if available, methanol-soluble filter measurements above the reported limit of detection. Regions identified for analysis are shown in black boxes and labelled. "Temperate/Boreal NA" refers to Temperate and Boreal North America, while "East of NZ" refers to the region southeast of New Zealand.



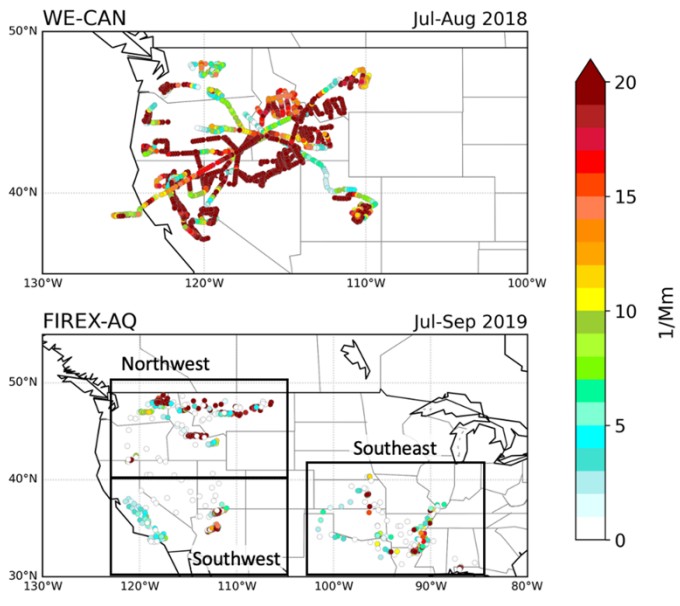

**Figure 2.** Map of BrC absorption coefficient measurements, at 365 nm, collected during the WE-CAN and FIREX-AQ flight campaigns. Total ambient absorption for all campaigns except WE-CAN above the reported limit of detection. WE-CAN ambient absorption is calculated from online WS absorption measurements, while FIREX-AQ ambient absorption is calculated from water soluble (WS) and, if available, methanol-soluble filter measurements. Regions identified for analysis are shown in black boxes and labelled.

Campaign data shown in Figs. 1 and 2 are grouped into labelled-regions for narrowed analysis, with the exceptions of the DC3 and WE-CAN campaigns that took measurements over smaller areas. To avoid overlap, regions covered by both ATom-3 and ATom-4 are only examined in one campaign. Further, measurements from ATom flight tracks with limited temporal coverage (that fall into too few model, 30-minute timesteps), for instance the track along 120ºW, are excluded.

## 2.2 Model description and experiments

### 2.2.1 The GISS ModelE Earth System Model

We used version 2.1 of the GISS ModelE Earth system model ("ModelE") (Kelley et al., 2020), with a horizontal resolution of 2º latitude by 2.5º longitude and 40 vertical layers from the surface to 0.1 hPa, for this study. For aerosol representation, we used the One-Moment Aerosol (OMA) module (Bauer et al., 2020), which is fully interactive within ModelE in terms of emissions, transport, removal, climate, and chemistry. OMA is a mass-based scheme where aerosols are assumed to have prescribed and constant size distributions. Represented aerosol components are sulfate, nitrate, ammonium, dust, sea salt, and carbonaceous aerosols. The latter consist of BC and OAs from both non-BB anthropogenic and BB sources. Biogenic SOA formation is simulated with a two-product model, producing aerosols from isoprene and α-pinene gases. These are the only sources of SOA in ModelE, as aromatic gases which would produce aromatic SOA are not explicitly represented. Sea





salt, dimethyl sulfide (leading to methanesulfonic acid), isoprene, and dust emission fluxes are calculated interactively, while
all other emissions (including BB fluxes) are prescribed. OA emissions inputs are in units of mass of organic carbon, while
ModelE tracks OAs and provides output in units of total organic aerosol mass. To convert between these, ModelE uses an
organic aerosol-to-organic carbon (OA-to-OC) ratio of 1.4:1. This OA-to-OC ratio value is consistent with OA schemes used
in other models (Tsigaridis et al., 2014).

OMA also includes a BrC aerosol scheme (DeLessio et al., 2024a). Four key properties, or processes, are defined to
simulate BrC aerosols: BB emissions, formation of secondary BrC, optical properties of primary and secondary BrC, and
chemical aging of primary BrC. Firstly, a BB BrC-to-OA emission proportion of 35% is prescribed, allocating that portion of
BB OA emissions as BrC, and leaving the remaining portion non-absorbing. Secondary BrC was accounted for by defining
biogenic SOA species as weakly absorbing, with imaginary RIs ≤ 0.002 in the model UV-VIS band. A primary BrC
imaginary RI of 0.0165 was set for the same band. Since BrC demonstrates limited absorption past 800 nm (Laskin et al.,
2015), BrC optical properties were only modified in the UV-VIS band; the optical properties of BrC in all other bands are
the same as non-absorbing OAs. Finally, an oxidant-concentration-driven aging scheme simulates primary BrC aging: the
heterogenous browning and bleaching of BrC is presented as mass transferred between BrC species with imaginary RIs
proportionally higher (more-absorbing) or lower (less-absorbing) than emitted BrC. Together, these properties make up the
"base case" of BrC representation.

More details about this scheme, including how each of the four properties/processes was estimated, can be found in
DeLessio et al. (2024a). These parameters are based on laboratory and field studies, but they naturally use assumptions and
simplifications, like the fact that they are constant throughout a simulation. One important limitation to note is that this
scheme does not include aqueous-phase or in-cloud production of BrC: while there are limited observations, some studies
have suggested this could be an important source of BrC at higher altitudes (Zhang et al., 2017, 2020; Kuang et al., 2024;
Wu et al., 2024).

Aerosol-radiation interactions (ARIs) and aerosol-cloud interactions (ACIs) are calculated within the radiation and cloud
schemes. Mie scattering is used to compute size-dependent scattering properties of clouds and aerosols, with wet aerosol size
and complex RI as inputs. Apart from swelling with water and coating of mineral dust, all OMA aerosols are treated as
externally mixed. Optical properties, and therein ARIs, are computed for 6 wavelength bands in the shortwave (SW) and 33
in the longwave (LW) (Bauer et al., 2010). As mentioned previously, the UV-visible (UV-VIS) radiation band (1 of 6 SW
bands) extends from 300 to 770 nm, so spectrally-weighted optical properties for the entire band are approximately
equivalent to properties at 550 nm. OMA also includes the first indirect effect of aerosol-cloud interactions (ACIs; Bauer et
al., 2020).

**2.2.2 Simulations for evaluation**



Nudged, transient model simulations were used for comparison against campaign data. After a five-year spin-up, 30-minute output was produced for each day covered by each flight campaign. 30-minutes is the highest temporal resolution available, as that is the length of a single ModelE time step. The specific dates for these periods of high temporal resolution model output can be seen in Appendix A (Table A1). BB emissions, including BrC, BC, and total OAs, were prescribed by the Global Fire Assimilation System version 1.2 (GFAS; Kaiser et al., 2012). GFAS was used, instead of other fire emission

inventories, because it contains information of plume injection height in each grid cell rather than the model default of emissions injected uniformly in the boundary layer. Since GFAS provides daily emissions, ModelE divides those by the number of time steps in a UTC-day to generate constant emissions for each 30-minute time step. This must be kept in mind in evaluating the model, as real-time fire emissions sampled by flights are not constant over an entire day. All other prescribed fluxes (referenced in Sect. 2.2.1) are from the Community Emissions Data System (CEDS; Hoesly et al., 2018).

These simulations were nudged towards 3-hourly winds provided by Modern-Era Retrospective Analysis for Research and Applications, version 2 (MERRA2; Gelaro et al., 2017). We use the base-case of BrC representation in simulations as a starting point for model comparison, but also run simulations with changes to individual OA or BrC properties for additional sensitivity tests (see Sect. 3.3 for more details).

To "sample" the model for comparison against flight data, we first determine the UTC time-step and grid cell each

campaign datapoint falls into, based on merged dataset time and location information. We then pull out the model output from that time step and cell, so it can be compared to the corresponding datapoint. Since flight measurements are typically taken at resolutions shorter than 30 minutes, datapoints that are in the same grid cell and same 30-minute period are averaged together. The spatial resolution of ModelE poses a limitation for exact comparison to campaign measurements: while a flight samples aerosol mass and absorption of a fire plume or, more generally, an inhomogeneous air mass at points along its flight

track, the same sample source, if captured by ModelE, is averaged over an entire grid cell. Thus, ModelE can't capture the same magnitude of aerosol properties for aerosol plumes not representative of a broad region, even if all emissions and processes are correct, inherently biasing the model low. To try to resolve this, we multiply model output of BrC, OA, and BC by a CO scaling factor, following Eq. (1):

$$\text{Scaling factor} = \frac{[\text{Campaign CO}]}{[\text{ModelE CO}]} \, . \tag{1}$$

As fires are a major source of CO, and it has a long atmospheric lifetime, it can be used as a tracer of fires (van der Werf et al., 2017). So, applying such a factor should account for the inability of the model to capture the same magnitude of emissions within the same grid cell. This does not, however, address the potential low bias from emissions or plumes transported in the model falling in an adjacent grid cell and not being sampled.

ModelE's calculation of optical properties in radiation bands, rather than distinct wavelengths, must also be addressed to

aid flight comparisons. As mentioned in Sect. 2.2.1, radiation calculations (relevant to BrC) are done over UV-VIS band, making model BrC absorption coefficient indicative of 550 nm. Since most campaigns report BrC absorption at 365 nm (see





Sect 2.1) we use this wavelength for evaluation, and an AAE value must be used to convert ModelE absorption from 550 to 365 nm. When measured BrC absorption is available at multiple wavelengths (in ATom and FIREX-AQ campaigns), we calculate AAE for each datapoint. When this isn't available, we assume a UV-VIS AAE value of 5.25, consistent with the

average of a uniform $AAE_{BrC}$ distribution from 4.1 to 6.4, as discussed in Zhang et al. (2017). Just as the imaginary RI of BrC varies with combustion source and processing, $AAE_{BrC}$ likely varies in the atmosphere (Forrister et al., 2015; Sumlin et al., 2018), so assuming one value for this introduces additional uncertainty to our analysis.

## 3 Results and Discussion

### 3.1 Background biases in BC comparisons

To evaluate model simulations of aerosols, we bin each campaign measurement, along with ModelE data sampled in the same model timestep and grid cell (see Sect. 2.2.2), into 500 m altitude bins ranging from the surface to 14 km. These datapoints are then averaged over the entire campaign period and the horizontal extent of the identified region of analysis (or the entire measurement area, as is the case for DC3 and WE-CAN campaigns). The result is a vertical profile, showing horizontally- and temporally-averaged aerosol concentration, or absorption, measured in-situ and simulated by ModelE, as a

function of altitude. The focus of this study is ModelE's BrC and OA scheme, specifically their simulation of BB-influenced airmasses. But because BC is often co-emitted with BrC and OAs from BB (Lack et al., 2012; Saleh et al., 2014; Pokhrel et al., 2016) it is helpful to first look at the performance of ModelE's BC simulation: if there are biases in model BC concentrations, subsequent biases seen in BrC could be a result of general aerosol processing or BB emissions, rather than specific improvements needed for BrC and OA representation.

We present vertical profiles of BC mass concentration over land, or in regions assumed to be near BB sources, in Fig. 3. Figure 4, on the other hand, shows BC profiles in more remote regions. In each of these figures, model-simulated BC is compared to campaign measurements, which in turn have an uncertainty between 20-30% (see Table 1). This is important to keep in mind, as model evaluation against these measurements will have some inherent uncertainty. Points at which there are BrC absorption (BrC-Abs) outliers, defined as BrC-Abs greater than two standard deviations above the campaign mean, have

been removed from consideration and, as mentioned previously, ModelE simulated data are multiplied by a CO-scaling factor. The vertical profiles of campaign and ModelE CO, for near-source and remote regions, are provided in Appendix A for reference (see Figs. A1-2).







**Figure 3.** Vertical profiles of BC mass concentration measured by flight campaigns (black) and simulated by ModelE's base case of BrC
representation (red), over land/near sources, reported in units of ng m$^{-3}$. Model data has been multiplied by a CO scaling factor (Eq. 1) and
is plotted at the mid-point of each altitude bin. Campaign and specific region of analysis is indicated on the top left of each plot. Dashed
lines indicate altitudes with no data. Shaded areas show variability of BC mass in each altitude bin, blue bars show the number of
datapoints averaged in each bin. Horizontal blue dashed lines separate profiles into the lower-troposphere (0-4 km), the mid-troposphere



(4-8 km), and the upper-troposphere (8-14 km). Outlier data (BrC-Abs greater than two standard deviations above the campaign mean)
have been removed from these average profiles. Data from DC3 and SEAC⁴RS campaigns have been further filtered for datapoints with
BB aerosol number fraction greater than 0.5 to narrow analysis to BB-influenced samples. Note: this BB fraction filter was not applied to
ATom, despite it being measured in that campaign, because it would reduce the already limited amount of data for analysis.

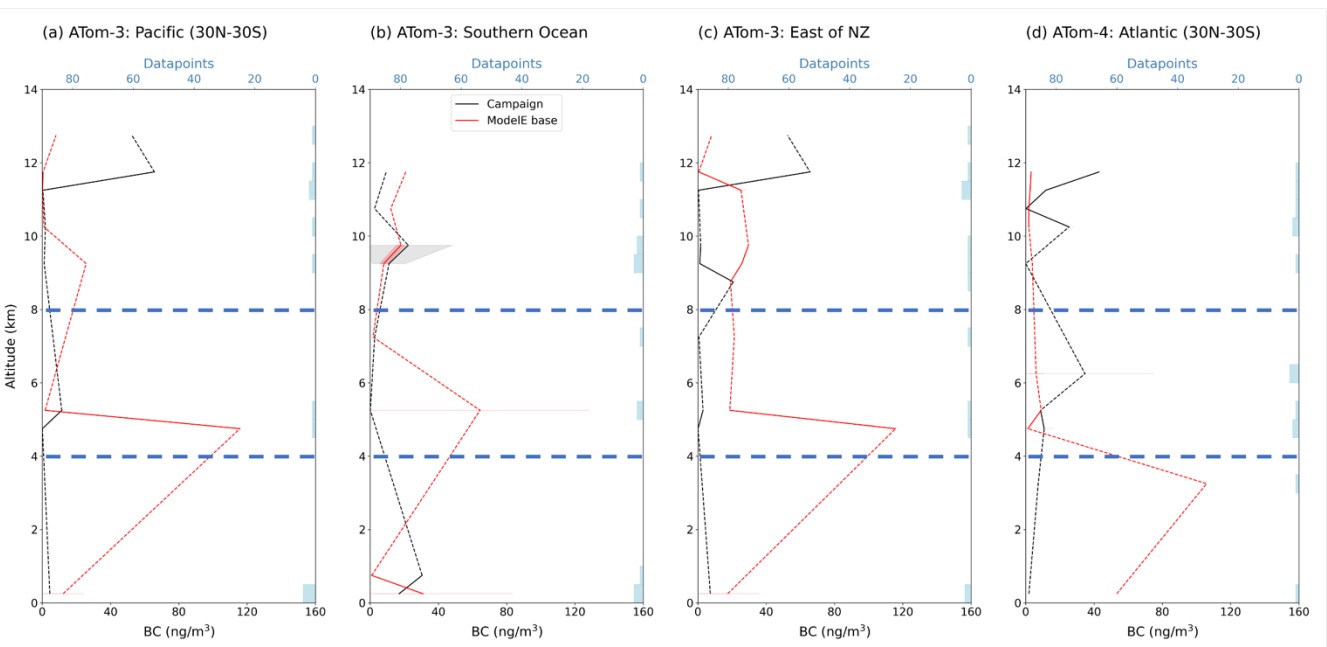

**Figure 4.** Vertical profiles of BC mass concentration, measured by flight campaigns (black) and simulated by ModelE's base case of BrC
representation (red), in remote regions, reported in units of ng m⁻³. Consistent with Fig. 3, a CO scaling factor (Eq. 1) has been applied to
model data, the midpoint of altitude bins is plotted, campaign and specific region of analysis is indicated on the top left of each plot,
dashed lines indicate altitudes with no data, shaded areas show variability of BC in each altitude bin, blue bars show the number of
datapoints averaged in each bin, horizontal blue dashed-lines separate the lower-, mid-, and upper-troposphere, and data is filtered for BrC-
Abs outliers. Unlike Fig. 3, there is no BB number fraction filter applied, as data is already limited.

Looking at the near-source regions in Fig. 3, there is low model bias in the lower-troposphere, with ModelE capturing or
slightly overestimating BC mass, except in the case of northwest and southeast FIREX-AQ (Fig. 2f,h). In these regions, the
model appears to underestimate BC near the surface. As mentioned previously, there is some negative bias due to dilution:
any fire plume sampled by FIREX-AQ would be averaged out over an entire grid-cell in ModelE, inherently decreasing
simulated BC magnitude. However, although the WE-CAN campaign also sampled aerosols directly from fire plumes,
ModelE appears to align well with WE-CAN-measured BC mass in the lower-troposphere. Another possible cause of
underestimated BC mass near the surface is low, or even missing, BB aerosol emissions over the U.S. during the FIREX-AQ
campaign period (July-September 2019). Since the model's BB emissions are prescribed by GFAS2.1, we don't explore this
possible bias further, as the evaluation of a fire emissions inventory is beyond the scope of this work.




In the mid-troposphere, ModelE slightly underestimates BC concentrations in most near-source regions (Fig. 3b-c,e-h).
There are several possible explanations for such negative biases at these altitudes. Low BB emissions, discussed previously, could bias aerosol mass low in the entire air column. However, since model BC is close to campaign data across most campaigns in the lower-troposphere of near-source regions, emissions is likely not a systematic cause of bias. Transport is another possible source of bias here: if ModelE doesn't simulate the vertical or horizontal transport of plumes or air masses as they occurred in the atmosphere, our analysis would show negative biases even if the correct magnitudes of BC mass were simulated. As ModelE is an Eulerian ESM, we quantify this potential bias. Finally, excess removal could contribute to low model BC in the mid-troposphere. A case study about the effect of cloud layers on tropospheric aerosols during SEAC⁴RS noted that layers of altocumulus (Ac) clouds in the mid-troposphere can cause the detrainment of aerosols, followed by microphysical processing and scavenging (Reid et al., 2019). It is possible that ModelE simulates too much scavenging, or too many clouds, and therefore removal of BC aerosols in such detrainment layers, creating a negative bias. We cannot definitively state this, though, as studying cloud cover or the incidence of clouds at this altitude was beyond the scope of this work.

This negative BC bias isn't observed in the mid-troposphere with DC3 and in the ATom-4 temperate and boreal North America region (Fig. 3a,d). These campaigns have the lowest BC concentrations in near-source regions, suggesting ModelE is less able to capture mid-troposphere BC mass at higher magnitudes. While there is more limited data, results in near-source upper troposphere follow a similar pattern as the mid-troposphere. In the lower and mid-troposphere over remote regions (Fig. 4), the model tends to overestimate low magnitude BC mass in ATom-3 Pacific, Southern Ocean, and East of NZ remote regions (Fig. 4a-c), again suggesting negative bias in ModelE doesn't occur with small BC concentrations. BC is similarly overestimated in the lower troposphere of the ATom-4 mid-Atlantic region but is underestimated in the mid-troposphere (Fig. 4d). This could indicate biases in BC transport or sinks as air masses move over the Atlantic, but it's difficult to draw meaningful conclusions from this, as these regional analyses are more limited in data and show much smaller magnitudes of measured BC mass than that of Fig. 3.

Through this evaluation of model-simulated BC mass concentration, we've identified potential sources of bias–low emissions, imperfect transport, excess scavenging–that are applicable to BB aerosols in general. To evaluate ModelE's BrC and OA schemes (the purpose of this study) we next need to determine if there is specific bias in model OAs, in addition to biases seen by BC aerosols. Figure 5 shows a scatterplot of BC versus total OA mass concentrations, for campaigns and ModelE.





**Figure 5.** Total OA concentration plotted against BC concentration, both in units of μg m$^{-3}$, as measured by flight campaigns (black x's) and simulated by ModelE (red o's). For each campaign (labelled on the top-left of each plot), data has been filtered to remove points outside of the regions used for vertical profile analysis and points coinciding with BrC-Abs outliers. Linear regression lines are included, and the slope and r$^2$ of these regressions are displayed on the top-right of each plot.

This shows that in most near-source regions (Fig. 5a-b,f), ModelE underestimates the ratio of OAs to BC. In ATom-3, the model seems to overestimate OAs relative to BC, but is closer to the OA-to-BC relationship in ATom-4 and WE-CAN measurements. It should be noted that these observed relationships are not definitive, as campaign OA concentration measurements have an uncertainty of 38% (see Table 1), in addition to BC measurement uncertainty. Nonetheless, these results suggest that, in some regions, there is bias specific to ModelE organics. To explore this further and remove the general BB aerosol biases seen in BC from consideration, we applied an additional scaling, or correction, factor to model output of BrC and OAs. The factor is calculated following Eq. (2):



$$\text{Additional scaling factor} = \frac{[\text{Campaign BC}]}{[\text{ModelE BC}]} \ . \tag{2}$$

By multiplying model output by this factor, in addition to the CO-scaling factor (see Eq. 1), we can look at BrC scheme performance in the hypothetical scenario that model BC and CO are completely correct. This does not mean related components, like the model's ability to capture a fire and transport its plume, are unbiased, but that these biases are not the focus of our study.

**3.2 Evaluation of ModelE BrC absorption**

Figures 6 and 7 show vertical profile comparisons of BrC absorption at 365 nm (BrC-Abs) measured in-situ and simulated by ModelE (with CO and BC scaling factors applied), in near-source and remote regions, respectively. It is important to remember that campaign measured absorption has its own uncertainty, with WS-Abs uncertainty between 12-18% and MS-Abs uncertainty between 15-21% (see Table 1). Therefore, the following comparisons cannot be considered definitive, but they do utilize the best available measurements for evaluation. Vertical profiles of total OAs and WSOC mass concentrations, with the same scaling factors applied, are provided in Figs. A3-6, for reference.





**Figure 6.** Vertical profiles of BrC absorption (BrC-Abs) at 365 nm, measured by flight campaigns (black) and simulated by ModelE's base case of BrC representation (red), over land/near sources, reported in units of Mm$^{-1}$. Model absorption has been multiplied by a CO and BC scaling factor (Eqs. 1-2). Consistent with Figs. 3 and 4, campaign and specific region of analysis is indicated on the top left of each plot, the midpoint of altitude bins in plotted, dashed lines indicate altitudes with no data, shaded areas show variability of data (here, BrC-



Abs) in each altitude bin, blue bars show the number of datapoints averaged in each bin, horizontal blue dashed-lines separate the lower-, mid-, and upper-troposphere, and data is filtered for BrC-Abs outliers. It is also filtered for BB number fraction (when relevant).

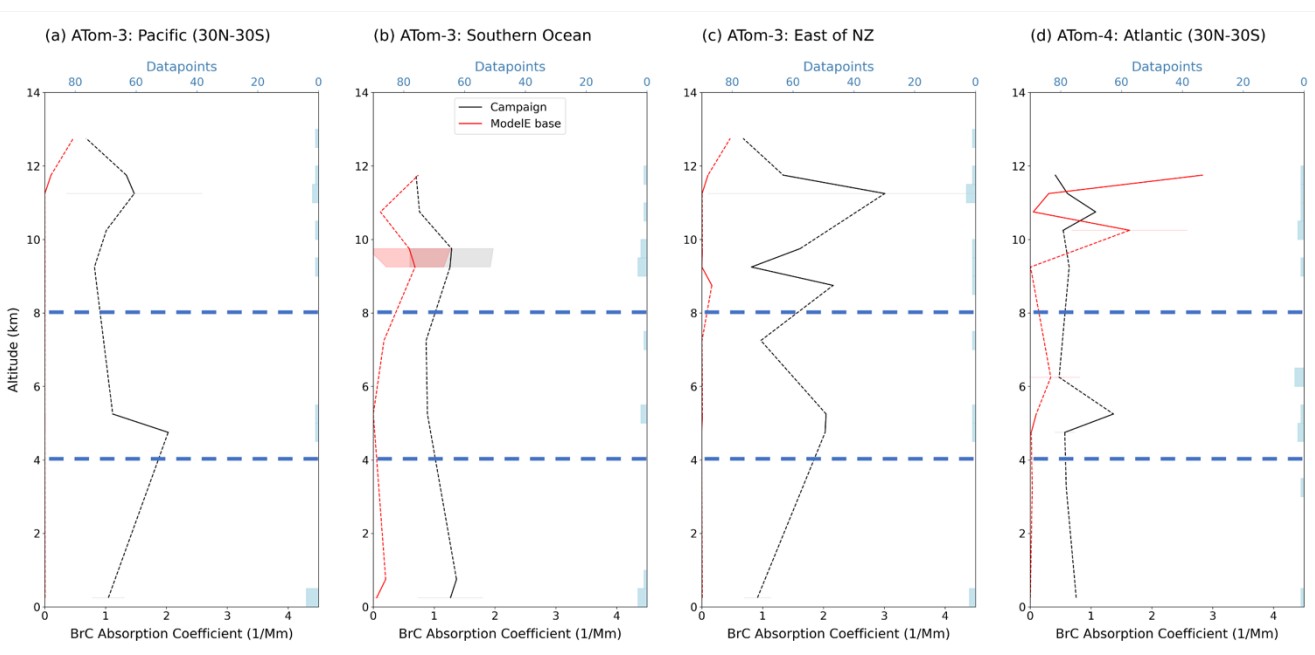

**Figure 7.** Vertical profiles of BrC absorption (BrC-Abs) at 365 nm, measured by flight campaigns (black) and simulated by (red), in remote regions, reported in units of Mm$^{-1}$. Consistent with Fig. 6, a CO and BC scaling factor (Eqs. 1-2) have been applied to model data, campaign and specific region of analysis is indicated on the top left of each plot, the midpoint of each altitude bin is plotted, dashed lines indicate altitudes with no data, shaded areas show variability of BrC-Abs in each altitude bin, blue bars show the number of datapoints averaged in each bin, horizontal blue dashed-lines separate the lower-, mid-, and upper-troposphere, and data is filtered for BrC-Abs outliers. Data is not filtered for BB number fraction.

ModelE appears to underestimate BrC-Abs across most regions and altitudes. To confirm this, we calculated the relative differences between model-simulated and campaign-measured BrC-Abs, following to Eq. (3):

$$\text{Relative difference} = \frac{\text{Abs}_{\text{ModelE}} - \text{Abs}_{\text{Campaign}}}{|\text{Abs}_{\text{Campaign}}|} \ . \tag{3}$$

Table 2 provides average relative differences for the lower-, mid- and upper troposphere as well as the entire atmospheric column, across all campaigns and sub-regions of analysis.

**Table 2.** Average BrC absorption relative difference (calculated according to Eq. 3) for each campaign and sub-region of analysis. Averages are provided for the lower (0-4 km), mid- (4-8 km), and upper troposphere (8-14 km), as well as the entire atmospheric column. Data is not available in the upper troposphere of the SEAC$^4$RS Southeast region, or during the WE-CAN campaign. Campaigns and sub-regions are listed in the same order they appear in vertical profile analyses for near-source (Figs. 3 and 6) and remote (Figs. 4 and 7)



regions. "Temperate/Boreal NA" refers to Temperate and Boreal North America, while "East of NZ" refers to the region southeast of New Zealand.

| Campaign and region of analysis | Avg. lower troposphere relative difference | Avg. mid-troposphere relative difference | Avg. upper troposphere relative difference | Avg. relative difference for entire column |
|---|---|---|---|---|
| DC3 | -0.82 | -0.50 | -0.35 | -0.52 |
| SEAC⁴RS West | -0.71 | -0.83 | -0.48 | -0.70 |
| SEAC⁴RS Southeast | -0.23 | -0.82 | -0.80 | -0.50 |
| ATom-4 Temperate/Boreal NA | -0.69 | -0.89 | -0.43 | -0.63 |
| WE-CAN | -0.29 | -0.68 | -0.21 | -0.46 |
| FIREX-AQ Northwest | -0.46 | -0.74 | -0.17 | -0.42 |
| FIREX-AQ Southwest | -0.82 | -0.87 | -0.83 | -0.84 |
| FIREX-AQ Southeast | -0.58 | -0.78 | -0.03 | -0.55 |
| ATom-3 Pacific | -1.00 | -1.00 | -0.85 | -0.90 |
| ATom-3 Southern Ocean | -0.90 | -0.90 | -0.45 | -0.67 |
| ATom-3 East of NZ | -0.99 | -1.00 | -0.86 | -0.91 |
| ATom-4 Atlantic | -0.97 | -0.73 | 1.09 | 0.13 |

While there appears to be a nearly systematic underestimation of BrC absorption, the relative magnitudes differ, and there are some instances when model absorption exceeds that of the campaign.

Looking first at absorption over near-source regions (Fig. 6), biases in the lower troposphere are not consistent across all campaigns. BrC-Abs appears to be more underestimated during DC3, in the SEAC⁴RS western region and the FIREX-AQ southeast regions (Fig.6a-b,g), while model absorption appears closer to measurements in the SEAC⁴RS southeast region, ATom-4 temperate/boreal North America region, and FIREX-AQ northwest and southeast regions, as well as during WE-CAN, and in the (Fig. 6c-f,h). OA comparisons (Fig. A3) in the same campaigns and regions generally follow similar patterns in the lower troposphere: where BrC-Abs is more underestimated, model OA mass is also low, but where model absorption is closer to measurements, OAs align well with or are overestimated compared to campaign data. Near-source WSOC mass (Fig. A5), while more variable in the lower troposphere, also follow a similar pattern (except for the SEAC⁴RS southeast region in which WSOC is more underestimated). This could suggest general bias in OAs, not specific to BrC. One potential cause of such bias is BB OA emissions being too low during certain campaign periods, as emissions influence the lower troposphere of all three of these variables.





405       There is an apparent underestimation in BrC-Abs in the mid-troposphere in most near-source regions. This is most apparent during the SEAC[4]RS campaign (Fig. 6b-c), and least during the DC3 campaign (with a smaller average relative difference of -0.5; Fig 6a). DC3 measured lower BrC-Abs, compared to some other near source regions, suggesting there could be less near-source model bias with smaller magnitudes of BrC-Abs. OAs and WSOC do not show the same consistent pattern of underestimation at these altitudes: model OA and WSOC mass is either less underestimated of well-aligned with

measurements. Finally, where there is data available, absorption is also underestimated in the upper troposphere of most near-source regions (Fig. 6a,c-f,h), but the relative magnitude of this negative bias is smaller than in the mid-troposphere. Model OAs and WSOC at these altitudes and in these regions are either closer to or greater than campaign measurements. There are two exceptions to this pattern in the upper troposphere: in the SEAC[4]RS west and FIREX-AQ southwest regions, there are large negative biases (similar in relative magnitude to the mid-troposphere; Fig. 6b,g). In these regions, model OAs

show limited bias in the upper troposphere, but simulated WSOC is also distinctly underestimated. As such, this bias could be result of diminished vertical transport of WSOC, and therein a portion of BrC, in these regions during campaign periods, though this cannot be definitively concluded.

      Vertical profile comparisons in more remote regions (Fig. 7) support the results shown in Fig. 6. Model absorption is typically underestimated in the lower and mid-troposphere across these regions, with some convergence in the upper

troposphere (particularly in the Atlantic during ATom-4; Fig. 7d) Also consistent with near-source mid- and upper troposphere results, OAs and WSOC in remote regions tend to be closer to or greater than campaign data (the latter is particularly true for OAs). Figs. 6 and 7, considered together, suggest that there may be biases specific to BrC in aged air masses and fire plumes (at higher altitudes and over remote regions).

      To further investigate this, we looked at BrC-Abs versus total OA mass concentration across all campaigns, as shown in

Fig. 8.







**Figure 8.** BrC absorption coefficient, in units of Mm$^{-1}$, plotted against total OA concentration, in units of µg m$^{-3}$, as measured by flight campaigns (black x's) and simulated by ModelE (red o's). For each campaign (labelled on the top-left of each plot), data has been filtered to remove points outside of the regions used for vertical profile analysis and points coinciding with BrC-Abs outliers. The slope and r$^2$ of

linear regressions for measured and simulated data are displayed on the top-right of each plot. Regression lines are plotted for all campaigns except SEAC$^4$RS (as there are two distinct branches of data) and ATom-3 (as the r$^2$ is too low).

Figure 8 suggests ModelE is underestimating BrC absorption, relative to OA mass, compared to all flight campaigns. The most significant bias appears with WE-CAN and FIREX-AQ, both of which sampled fresh BB plumes. This negative bias is evident even in data from DC3 and SEAC$^4$RS: despite some spread, the majority of datapoints appear to fall above the

model. These results suggest that ModelE's scheme is either simulating too little BrC mass, or the BrC mass present is not absorbing enough.

### 3.3 Designing sensitivity tests for potential sources of bias





In the previous section, we identified a pattern of underestimation in model BrC-Abs that could be driven by biases with model OAs (in the lower troposphere) as well as biases specific to the BrC scheme (in aged air masses and fire plumes). The
next step is to identify specific mechanisms that could be driving these biases, then determine changes to ModelE BrC and OA schemes to test these potential bias sources. We introduce and provide rationale for several model sensitivity tests in this section, then present the results of these tests in Sect. 3.4.

Low BB OA emissions were identified as a possible explanation for the underestimation of BrC-Abs (as well as OAs and WSOC) seen in the near-source lower troposphere during some campaigns (Fig. 6a-b,g), and seen to a lesser degree during
others (Fig. 6c-f,h). To address this, we ran a sensitivity test that uniformly scaled up global BB OA emissions by 50%. The low bias could also be driven by the OA-to-OC ratio used for ModelE's OAs, mentioned in Sect. 2.2.1. If the model's constant value of 1.4 is too low, then OA emissions, and therefore BB BrC mass, would also be biased low. Since the DC3, SEAC[4]RS, ATom, and WE-CAN campaigns provided both OA and OC mass measurements, we can directly evaluate our value against in-situ data. We applied a linear regression to campaign data of OA and OC mass, using the datapoints sampled
for vertical profile analysis, to calculate average OA-to-OC ratios for each flight campaign (taken from the slope of the linear regression). The average DC3 ratio was 2.02, SEAC[4]RS was 1.99, ATom-3 was 2.10, ATom-4 was 2.20, and WE-CAN was 1.80. This confirms the prescribed ModelE ratio is too low, particularly for campaigns sampling more aged OAs in remote regions. Laboratory and field studies of OAs have also suggested the traditional model value of 1.4 may be too low: aged organics have been found to have higher OA-to-OC ratios, and while 1.4 may be appropriate for urban and industrial OAs
(with a suggested range of 1.4-1.6), oxidized organics have ratios around 2.0, and BB OAs are somewhere in the middle (1.6-1.8) (Aiken et al., 2008; Philip et al., 2014; Tsigaridis et al., 2014; Canagaratna et al., 2015; Chrit et al., 2018; Zhu et al., 2023). Canagaratna et al. (2015) reported chamber SOA ratios to be as high as 2.3.

As flight campaign data suggest the ModelE OA-to-OC ratio value of 1.4 is too low, and previous literature reports different values for different types of OAs, we assigned new values to some of the OA species in the model. The specific
values are listed in Table 3, but, in general, the fresh industrial OA ratio remained the same, while the ratios of BB OAs, all aged OAs, and SOAs were increased.

**Table 3.** Updated values of prescribed organic aerosol-to-organic carbon (OA-to-OC) ratios used by ModelE organic aerosol species.

| ModelE organic aerosol type | Updated, prescribed OA-to-OC ratio |
|---|---|
| Fresh, industrial OAs | 1.4 |
| BB OAs (BrC and non-absorbing BB OAs) | 1.8 |
| Aged OAs (aged BrC and aged industrial OAs) | 2.0 |
| SOAs (from both isoprene and $\alpha$-pinene precursors) | 2.3 |



With these changes, we ran a simulation over campaign periods to test the efficacy of the new OA-to-OC ratios. Figure 9, a scatterplot of OC versus total OA mass concentration for each campaign and model simulation, shows the effect of varying

OA-to-OC ratios by species.



**Figure 9.** Total OA concentration plotted against OC concentration, both in units of µg m$^{-3}$, as measured by flight campaigns (black x's), simulated by the base case of ModelE OA representation (red o's), and simulated by the revised ModelE case with variable OA-to-OC ratios (purple o's). For each campaign that provided both OC and OA measurements (labelled on the top-left of each plot), data has been

filtered to remove points outside of the regions used for vertical profile analysis and points coinciding with BrC-Abs outliers. Linear regression lines for campaign data and all model simulations are included. The slope and r$^2$ of campaign and revised-case regressions are displayed on the top-right of each plot for comparison–these slopes are indicative of average OA-to-OC ratios. The slope and r$^2$ of the model base case is not provided because the value (1.4) is constant across all campaign periods and regions.



Varying the OA-to-OC ratio by OA species greatly improved model agreement with measured OC versus OA
concentrations in every flight campaign. These simple changes linearly increased OA sources, for instance BB OA emissions
increased by 28.5% (new ratio-old ratio/old ratio; 1.8-1.4/1.4=0.285), but they also near-linearly increased all sinks. Since
OA sinks occur across all latitudes and longitudes, while sources occur only in certain areas, the net effect of this change was
a decrease in global average OA burden. While these changes will likely not reduce the apparent negative bias in OA mass
and BrC-Abs, it is clearly more physically accurate than the base case. As such, these updated OA-to-OC ratios were used
for all sensitivity tests in this study.

Assuming general bias from transport has been accounted for with the BC scaling factor, underestimation bias specific to
BrC in aged air masses and fire plumes could be due to missing secondary sources or excess sinks. Secondary production of
BrC in fire plumes, or in clouds during convection, could be a source of bias as ModelE doesn't include either of these
processes. Starting with the former, laboratory and field studies have found that BBSOA could be an important source of
BrC (Saleh et al., 2013; Kumar et al., 2018; Saleh, 2020; Palm et al., 2020; Kuang et al., 2024). Such BBSOA production
could help explain observations from WE-CAN and FIREX-AQ: there was no clear evidence of BrC aging observed in fresh
plumes (approximately up to 9 hours in age) in either of these campaigns (Sullivan et al., 2022; Washenfelder et al., 2022).
One study of FIREX-AQ measurements reported no consistent pattern in BrC production or depletion, attributing this to the
likely complex evolution of BrC in smoke (Zeng et al., 2022). No change in BrC absorption in a plume could mean either the
BrC present is stable and not bleaching, or secondary production of BrC is balancing the aging and removal of primary BrC.
While we can't currently represent BBSOA in ModelE, a common issue across many models, we could limit the aging in our
BrC scheme to try to capture this pattern. Schnitzler et al. (2022) developed a kinetic model of BrC aging, via oxidation by
ozone, that considered relative humidity (RH) and temperature, and found that the BrC aging timescale of ~1 day employed
by several models is only applicable below the planetary boundary layer (PBL). They suggest that above the PBL, aging is
much slower due to low RH and temperature. Based on this study, we ran a sensitivity test that removed any aging for BrC
above the PBL. This change aims to both increase the physical correctness of the BrC scheme and reduce the bias of BrC-
Abs at higher altitudes.

In-cloud production could be another source of BrC our scheme isn't able to capture. Zhang et al. (2017) attributed
enhancements of BrC, relative to BC, observed in the mid- and upper-troposphere during DC3 to either secondary BrC from
in-cloud processing or enhanced convection of BrC. In a subsequent modelling study, the same group found that reducing
wet scavenging of BrC improved alignment of the Community Earth System Model (CESM) BrC scheme with both DC3
and SEAC[4]RS data (Zhang et al., 2020). Here, they reduced sinks of BrC, rather than adding an in-cloud source as there
wasn't enough observational data to inform the model development of such a mechanism. We could follow the same
approach: ModelE doesn't have an in-cloud OA chemistry mechanism in which BrC production could be implemented, but
we can reduce the fraction of BrC defined as water-soluble (WS). This would reduce wet removal of BrC, particularly in
clouds and convective systems, and could address the underestimation of BrC-Abs in the mid- and upper-troposphere.





In the base case of BrC representation, there are three primary BrC species–an emitted BrC, a browner BrC, and a bleached BrC–that all have the same prescribed WS fraction of 0.8 (DeLessio et al., 2024a). To decrease wet removal of BrC, we ran a sensitivity test that reduced the emitted BrC fraction to 0.6, the browner fraction to 0.5, and the bleached
fraction to 0.4. These values are more consistent with the range provided in literature (Laskin et al., 2015; Zeng et al., 2020), and allow for the WS fraction to decrease with age. The latter point is consistent with field measurements that have found methanol-soluble (MS) BrC-Abs decreases at a slower rate than WS BrC-Abs, meaning more aged BrC, with some recalcitrant absorption, would have a large water-insoluble fraction (Wong et al., 2019). We did not change the WS fractions of biogenic SOA, which are considered brown, from their constant value of 0.8. This value is not only commonly used by
other modeling groups (Tsigaridis et al., 2014), but it is also consistent with field measurements of SOA in the southeast U.S., a region dominated by biogenic SOA and focused on in this study (Xu et al., 2017).

Table 4 provides a summary of BrC and OA properties in each sensitivity test described here, as well as those of the BrC base case for reference. As stated previously, varied OA-to-OC ratios were found to increase physical correctness of the OA scheme, and thus used in all sensitivity tests.

**Table 4.** BrC and OA scheme properties for base case of representation and additional sensitivity tests.

| Simulation | Prescribed OA-to-OC ratios | BB OA emissions | Primary BrC aging | Prescribed WS fractions |
|---|---|---|---|---|
| Base | *All OAs:* 1.4 | 1x GFAS1.2 | All altitudes | *All BB OAs:* 0.8 |
| Var. OA:OC | | | | |
| Var. OA:OC/ 1.5x emis | | 1.5x GFAS 1.2 | | |
| Var. OA:OC/ lim. aging | *See values provided in Table 1* | 1x GFAS1.2 | No aging above PBL | |
| Var. OA:OC/ more WIBrC | | | All altitudes | *Emitted BrC:* 0.6 *Browner BrC:* 0.5 *Bleached BrC:* 0.4 |

**3.4 Sensitivity test results**

Figures 10 and 11 show the same vertical profile analysis, comparing flight campaign measured and ModelE simulated BrC-Abs in near-source and remote regions, as Figs. 6 and 7, but the results of additional sensitivity tests are included.







**Figure 10.** Vertical profiles of BrC absorption (BrC-Abs) at 365 nm, measured by flight campaigns (black) and simulated by ModelE's base case of BrC representation (red), variable OA-to-OC ratio case (purple), 1.5 times BB OA emissions case (blue), limited aging case (green), and larger fraction of WIBrC case (orange). Regions of analysis are over land/near sources, and data is in units of Mm$^{-1}$. Consistent with Fig. 6, a CO and BC scaling factor (Eqs. 1-2) have been applied to model data, campaign and specific region of analysis is indicated on the top left of each plot, the midpoint of altitude bins is plotted, dashed lines indicate altitudes with no data, shaded areas



show variability of BrC-Abs in each altitude bin, blue bars show the number of datapoints averaged in each bin, horizontal blue dashed-
lines separate the lower-, mid-, and upper-troposphere, and data is filtered for BrC-Abs outliers and BB number fraction (when relevant).

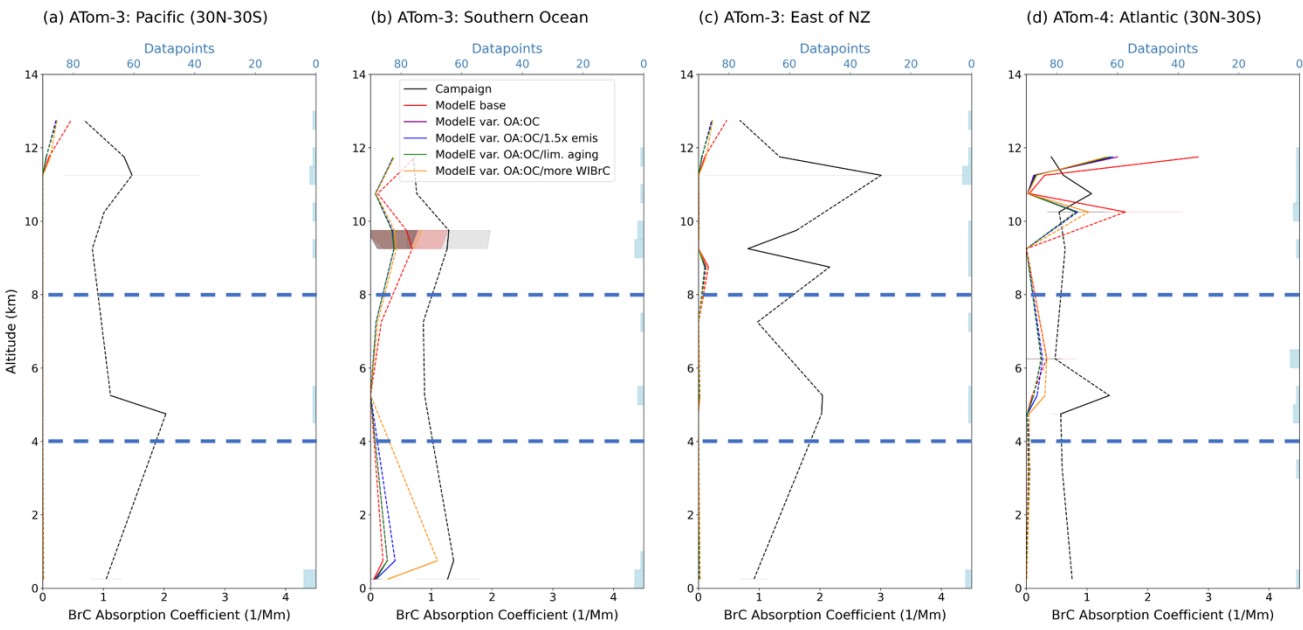

**Figure 11.** Vertical profiles of BrC absorption (BrC-Abs) at 365 nm, measured by flight campaigns (black) and simulated by ModelE's base case of BrC representation (red), variable OA-to-OC ratio case (purple), 1.5 times BB OA emissions case (blue), limited aging case

(green), and larger fraction of WIBrC case (orange). Analysis is over remote regions, and data is in units of Mm$^{-1}$. Consistent with Fig. 7, a CO and BC scaling factor (Eqs. 1-2) have been applied to model data, campaign and specific region of analysis is indicated on the top left of each plot, the midpoint of altitude bins is plotted, dashed lines indicate altitudes with no data, shaded areas show variability of BrC-Abs in each altitude bin, blue bars show the number of datapoints averaged in each bin, horizontal blue dashed-lines separate the lower-, mid-, and upper-troposphere, and data is filtered for BrC-Abs outliers. Data is not filtered for BB number fraction.

Looking at the lower-troposphere in near-source regions, there appears to be a small increase in BrC-Abs with the varied OA-to-OC, limited aging, and increased WIBrC cases during DC3, in the SEAC$^4$RS West region, in the ATom-4 temperate/boreal NA region, during WE-CAN, and in the FIREX-AQ north- and southwest regions (Fig 10a-b,d-g). In most of these regions, the effects of these cases are not distinguishable from each other; the increased WIBrC case shows a distinct absorption enhancement in the lower troposphere of the ATom-4 temperate/boreal North America and FIREX-AQ

northwest regions. In contrast to these results, all sensitivity tests showed a slight decrease in BrC-Abs below 2 km in the SEAC$^4$RS and FIREX-AQ southeast regions (Fig. 10c,h). This could be due to the increase in OA-to-OC ratios, particularly SOAs: biogenic SOAs, rather than BB OAs, tend to dominate in the southeast U.S., so a reduction in SOA burden because of this model change could have a greater impact in the PBL here.



Enhanced BB OA emissions increased lower troposphere BrC-Abs relative to other sensitivity tests across all near-source
regions except the ATom-4 temperate/boreal NA region, which sampled  air masses further from direct emission sources.
This sensitivity test brought the model closer to measured lower troposphere absorption in  most cases, excluding the
previously mentioned southeast regions (Fig. 9c,h), but WE-CAN is a notable exception to this (Fig. 10e). In the base case of
BrC representation, model BrC-Abs in the lower troposphere was close to WE-CAN data, so the uniform enhancement in
emissions led to ModelE overestimating absorption below 3 km. It's evident that while increased BB OA emissions can
improve model performance, it should not be applied uniformly without consideration for regional and temporal differences.

In the near-source mid-troposphere, all sensitivity tests showed enhanced BrC-Abs compared to the base case, again
except for U.S. southeast regions. The increased WIBrC case was distinguishable in more regions at these altitudes: in
regions and campaigns presumably influenced by larger fires (SEAC[4]Rs west, WE-CAN, and all FIREX-AQ regions; Fig.
10b,e-g), this test resulted in similar or greater BrC-Abs increases as the enhanced emissions case, with the relative impact
increasing with altitude. This supports the idea that excess wet removal of BrC could contribute to a negative bias in BrC
absorption in fire plumes. This effect can also be seen in more aged air masses, as in ATom-4's temperate/boreal NA region,
where more WIBrC led to the largest increases in mid-troposphere absorption. Despite absorption enhancements due to
increased emissions and WIBrC, there is still persistent underestimation of model BrC-Abs in the near-source mid-
troposphere. Where data is available, all sensitivity tests appear to simulate less absorption than the base case in the upper
troposphere of near-source regions. This is likely because all tests utilize the updated OA-to-OC ratios. Though these are
more physically correct, they reduce OA and BrC burden (see Sect. 3.3), and evidently exacerbate upper troposphere BrC-
Abs biases.

Sensitivity tests in remote regions generally support the results in near-source regions. Consistent with the mid-
troposphere of near-source regions, where remote region BrC-Abs was underestimated in the base case, that negative bias
persists across all sensitivity tests. Further, the same pattern of lower simulated BrC-Abs in the near-source upper
troposphere with sensitivity test cases, compared to the base case, is also seen in the upper and even mid- troposphere of
remote regions. Below the upper troposphere, the ATom-3 Pacific and east of NZ regions show little to no change with
sensitivity tests (Fig. 11a,c). As with the base case, it appears ModelE simply doesn't capture the BrC signal here. However,
more differentiation between sensitivity tests can be seen in the other two remote regions. Like the ATom-4 temperate/boreal
NA region, the increased WIBrC case showed the greatest increase in BrC-Abs in the mid-troposphere of the ATom-4
Atlantic region (Fig. 11d). This case also showed significant increases in BrC-Abs, relative to all other test cases, from the
surface to approximately 5 km in the ATom-3 Southern Ocean region (Fig. 11b). It appears that less wet removal of BrC
impacts the aged air masses in both regions.

In both near-source and remote regions, the limited aging sensitivity test is nearly indistinguishable from the varied OA-
to-OC ratio case; the test effect is only apparent in the upper troposphere of the FIREX-AQ southeast region, where it
simulated the least absorption of all test cases. The only difference between the aging and OA-to-OC cases is the limit of no



primary BrC aging above the PBL, so this scheme change had no evident effect. Average chemical lifetime of emitted BrC (global average over 2018, as an example) increased by only 0.13% and lifetime of browned BrC did not change, confirming the altitude-limit had no impact. It is likely that all primary BrC aging in ModelE takes place within the PBL, so removing

aging above that point does nothing–BrC has already browned then bleached. This suggests an additional limitation would need to be applied to the primary BrC aging scheme to investigate this potential bias.

In our current scheme, we simulate aging through mass transfer from one type of BrC to the next (emitted to browner, and browner to bleached, or threshold, BrC; DeLessio et al., 2024a). Rather than computing the rate of transfer between BrC species from second-order rate constants and local model oxidant concentrations (the current aging scheme), we could set a

fixed lifetime of 1 day (note: 15+ hr is the plume age at which FIREX-AQ data began to show BrC bleaching). Though this would reduce the chemical complexity of the BrC scheme, it could slow down aging, presumably improving model alignment with in-situ absorption measurements. Alternatively, Zhang et al. (2020) found that in addition to reducing BrC wet scavenging (noted in Sect. 3.3), limiting photobleaching to BrC outside of convective clouds improved model alignment with flight campaign data. Implementing such a limit in ModelE would first require the development of an in-cloud OA

chemistry scheme. BrC aging could then be differentiated, and limited, by cloud type. It bears mentioning that implementing additional sources of BrC, for instance production of BBSOA or in-cloud SOA, could help balance the effect of primary BrC aging on BrC-Abs evolution.

## 4 Conclusions

Representation of BrC aerosols, the subset of OAs that absorb light in the UV-to-visible wavelength range, in climate models

is necessary to constrain estimates of OA radiative forcing. This is especially true because OAs are becoming more prominent with increasing frequency and intensity of wildfires but estimates of their radiative forcing remain highly uncertain ($-0.21 \pm 0.23$ W m$^{-2}$; Szopa et al., 2021). In a previous study, a BrC scheme was implemented in the NASA GISS ModelE ESM (DeLessio et al., 2024a). While the addition of prognostic BrC improved the physical and chemical complexity of OAs in ModelE, using simplified parameterizations to represent highly variable observed BrC properties left

the scheme with many uncertainties. In this study, we used in-situ measurements of BrC absorption (BrC-Abs) from five different flight campaigns–DC3, SEAC[4]RS, ATom, WE-CAN, and FIREX-AQ–to evaluate and constrain ModelE's BrC scheme. Through vertical profile comparisons, we found a systematic underestimation of model BrC-Abs relative to campaign measurements. By correcting for biases in CO and BC, then examining the measured and simulated relationships between BrC and total OAs, we identified possible causes of this bias. Finally, we ran sensitivity tests to assess whether

model changes can reduce bias.

The first model change, varying OA-to-OC ratio by model OA species rather than using one value for all OAs, greatly improved model alignment to the same metric reported by flight campaigns. As these updated values improved the physical correctness of the ModelE OA scheme, and were supported by literature values, they were maintained in all additional tests



despite the resulting decrease in OA burden. Scaling up BB OA emissions by 50%, the next sensitivity test, improved model
performance in some near-source regions, but led to overestimation compared to the WE-CAN campaign. This suggests that
while increasing BB emissions can reduce model bias, particularly in the lower-troposphere near sources, it should not be
done uniformly over space and time. Remaining sensitivity tests aimed to address biases specific to BrC: underestimations in
absorption could be partially due to missing sources or excess sinks; we explored the latter. Increasing WIBrC fractions
resulted in distinguishable increases in mid-troposphere BrC-Abs over regions influenced by larger fires and some remote
regions, appearing to partially address the possible bias of excess BrC sinks in fire plumes and aged air masses. The
sensitivity test that limited aging of primary BrC to below the PBL, on the other hand, had limited effect on model data.

Despite implementing physically correct changes based on literature studies, like variable OA-to-OC ratios and more
reasonable WS BrC fractions, there are persistent biases between ModelE BrC absorption and in-situ measurements. Firstly,
there are general biases in BB aerosol processes, like emissions and transport, that fall outside the scope of our study. These
were removed from consideration to the greatest extent possible by applying both CO and BC scaling factors to BrC and OA
model output. In terms of remaining biases specific to the BrC scheme, increasing WIBrC fractions seemed to account for
some of the potential bias of excess sinks, but further bias due to aging of primary BrC could not be addressed. We can't
conclude this is because it doesn't cause bias in BrC absorption–the scheme changes made simply had no effect on model
output. Since this potential bias can't be eliminated, future work should implement more effective changes to the primary
BrC chemical aging scheme. Possible modifications, like adding a fixed lifetime or limiting aging to non-convective
transport, were discussed in Sect. 3.4. We were also unable to investigate missing secondary BrC sources as a cause of low
absorption bias. As discussed in Sect. 3.3, recent studies have highlighted the likely importance of BBSOA and in-cloud
production of SOA as sources of BrC, but ModelE doesn't currently simulate either SOA type. Like excess aging, we are
unable to rule these out as potential causes of bias. Representation of these two SOAs is a priority for OA scheme
development in ModelE. In the meantime, further laboratory and field studies of BBSOA/in-cloud SOA absorption are
needed to guide implementation of BrC into these SOA schemes, when model capability allows.

The results of this study are consistent with our previous work, where we evaluated ModelE against a BrC retrieval of
AERONET data. Comparisons of retrieved and simulated BrC AOD, AAOD, and mass suggested the model's base case of
representation didn't simulate enough absorption per unit mass (DeLessio et al., 2024b). This comparison also found that
changing BB emissions can improve model performance, but should be done with a region-specific approach, not uniformly.
Unlike the present work, however, we investigated and addressed biases by aligning model properties to retrieval
assumptions, which lead to less physically correct changes like eliminating primary BrC aging completely. Additionally, the
previous study focused on several different BB regions. The results we've presented here, however, are mostly indicative of
temperate and boreal North America, specifically the continental U.S. This is because the DC3, SEAC[4]RS, WE-CAN, and
FIREX-AQ campaigns operated out of and flew over the U.S., while the ATom campaign, which did sample a greater



variety of more remote regions, had limited data available for our analysis. More systematic measurements of BrC absorption are needed to constrain model representations on a global scale.

As mentioned in Sect. 2.2.2, there are limitations to evaluating an ESM like ModelE against flight campaign measurements. Comparing data collected at a single point in space and time to a model with 30-mintue timesteps and 2º by

2.5º grid cells is difficult. Sub 30-minute changes in measured BrC absorption are averaged out when converting to the model time resolution, and fire plumes or point sources of BrC become diluted when averaged over a model grid box. There are also limitations specific to BrC. An assumed water soluble-to-methanol soluble BrC ratio of 0.5, as well as an assumed bulk to ambient conversion factor of 2, were used to obtain ambient BrC absorption from in-situ filter measurements. When AAE couldn't be computed from campaign data, an assumed AAE of 5.25 was used to convert model absorption in the UV-

vis band (indicative of 550 nm) to absorption at 365 nm. Each of these assumptions were based on previous laboratory and field studies, but they still introduce uncertainty into our results. Finally, there are inherent uncertainties associated with in-situ measurements from flights, which we noted in Table 1 and referenced throughout this work. In this study, we focused on scheme parameters that could contribute to model biases in BrC absorption, taking measurements not as the definitive truth, but instead as the best available information for model evaluation. However, the sensitivity of the model-campaign

comparison to these uncertainties also warrants future consideration. Additionally, model-comparison sensitivity to variable water soluble-to-methanol soluble and bulk-to-ambient absorption conversion factors, as well as AAE should be explored. Further sensitivity tests like these could complement this work, investigating not just the skill of the current BrC scheme but also the efficacy of model evaluation against in-situ flight measurements.

Despite the limitations and uncertainties outlined above, we observed a consistent underestimation bias in ModelE BrC

absorption coefficient across five different flight campaigns. The results and subsequent discussion we presented here allowed us to highlight important processes of BrC, OAs, and BB aerosols in general, and to identify areas that require additional study.

*Code and data availability.* The GISS ModelE code is publicly available at https://simplex.giss.nasa.gov/snapshots/ (National Aeronautics and Space Administration, 2025). The most recent public version is E2.1.2. The Fortran code used for

the simulations described in this study, along with the model output, is available at https://doi.org/10.5281/zenodo.15092997 (DeLessio et al., 2025). The model code can be found in the file titled "ModelE_code_032625.tar.gz"; model output is in the file titled "ModelE_SimulationData.tar.gz". Model simulation data sampled at the same time and location of measurements is provided for each campaign. MERRA-2 reanalysis is available at https://doi.org/10.5067/QBZ6MG944HW0 (GMAO, 2015). DC3 campaign data are available at https://www-air.larc.nasa.gov/cgi-bin/ArcView/dc3 (NASA Airborne Science

Data for Atmospheric Composition, 2025a). SEAC⁴RS campaign data are available at https://www-air.larc.nasa.gov/cgi-bin/ArcView/seac4rs (NASA Airborne Science Data for Atmospheric Composition, 2025b). ATom campaign data are available at https://doi.org/10.3334/ORNLDAAC/1925 (Wofsy et al., 2021). WE-CAN and FIREX-AQ campaign merged



data are available at https://www-air.larc.nasa.gov/cgi-bin/ArcView/firexaq?MERGE=1 (NASA Airborne Science Data for Atmospheric Composition, 2025c). BrC absorption and WSOC mass concentration measurements from WE-CAN are not
included in merged data but can be found individually at https://doi.org/10.26023/CRHY-NDT9-C30V (Sullivan et al., 2021).

*Author contributions.* MAD, KT, and SEB conceived the study. All model development was done by MAD, guided by KT and SEB. MAD conducted all model simulations and analysis, created all figures, and drafted the first version of this manuscript. All authors contributed to later drafts.

*Competing Interests.* Kostas Tsigaridis is a member of the editorial board of Atmospheric Chemistry and Physics.

*Acknowledgements.* Climate modeling at GISS is supported by the NASA Modeling, Analysis, and Prediction Program. Maegan A. DeLessio acknowledges support from the Future Investigators in NASA Earth and Space Science and Technology program (grant no. 80NSSC22K1442). Kostas Tsigaridis acknowledges support from the Plankton, Aerosol, Cloud, ocean Ecosystem project (grant no. 80NSSC20M0205). Maegan A. DeLessio acknowledges useful guidance from
Róisín Commane and Faye McNeill. We thank Rodney Weber, the PI of BrC measurements taken in DC3, SEAC[4]RS, ATom, and FIREX-AQ flight campaigns, and Amy Sullivan, the PI of BrC measurements taken in the WE-CAN campaign. We thank Mary Barth/Chris Cantrell, Brian Toon, Steven Wofsy, Emily Fischer (and others), and Carsten Warnecke (and others) for their work as PIs of the DC3, SEAC[4]RS, ATom, WE-CAN, and FIREX-AQ campaigns, respectively. We thank Ru-Shan Gao, Joshua Schwarz, and Paul DeMott, the PIs of BC measurements. We thank Jose Jimenez, Lauren Garofalo,
and Delphine Farmer, the PIs of OA measurements. We thank Glenn Diskin, Kathryn McKain, and Colm Sweeney, the PIs of CO measurements. Finally, we thank Karl Froyd and Daniel Murphy, the PIs of DC3 and SEAC[4]RS PALMS data. Resources for this work were provided by the NASA High-End Computing (HEC) Program through the NASA Center for Climate Simulation (NCCS) at Goddard Space Flight Center.



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





**Appendix A:**

**Table A1.** Dates used for analysis, during which ModelE outputted simulated data in 30-minute timesteps, for each flight campaign studied.

| Campaign | Dates of analysis |
|----------|-------------------|
| DC3 | May 4-June 22, 2012 |
| SEAC$^4$RS | August 6-Septermber 23, 2013 |
| ATom-3 | September 28-October 27, 2017 |
| ATom-4 | April 24-May 21, 2018 |
| WE-CAN | July 24-September 13, 2018 |
| FIREX-AQ | July 22-September 5, 2019 |





**Figure A1.** Vertical profiles of carbon monoxide (CO) concentration measured by flight campaigns (black) and simulated by ModelE's base case of BrC representation (red), over land/near sources, reported in units of ppbV. Campaign and specific region of analysis is indicated on the top left of each plot, data is plotted at the midpoint of each altitude bin, dashed lines indicate altitudes with no data, shaded areas show variability of data in each altitude bin, blue bars show the number of datapoints averaged in each bin, horizontal blue dashed-lines separate the lower-, mid-, and upper-troposphere. To ensure these profiles represent the same sampled time and grid-cells as that of BrC-Abs, data is filtered for BrC-Abs outliers instead of CO. It is also filtered for BB number fraction (when relevant).





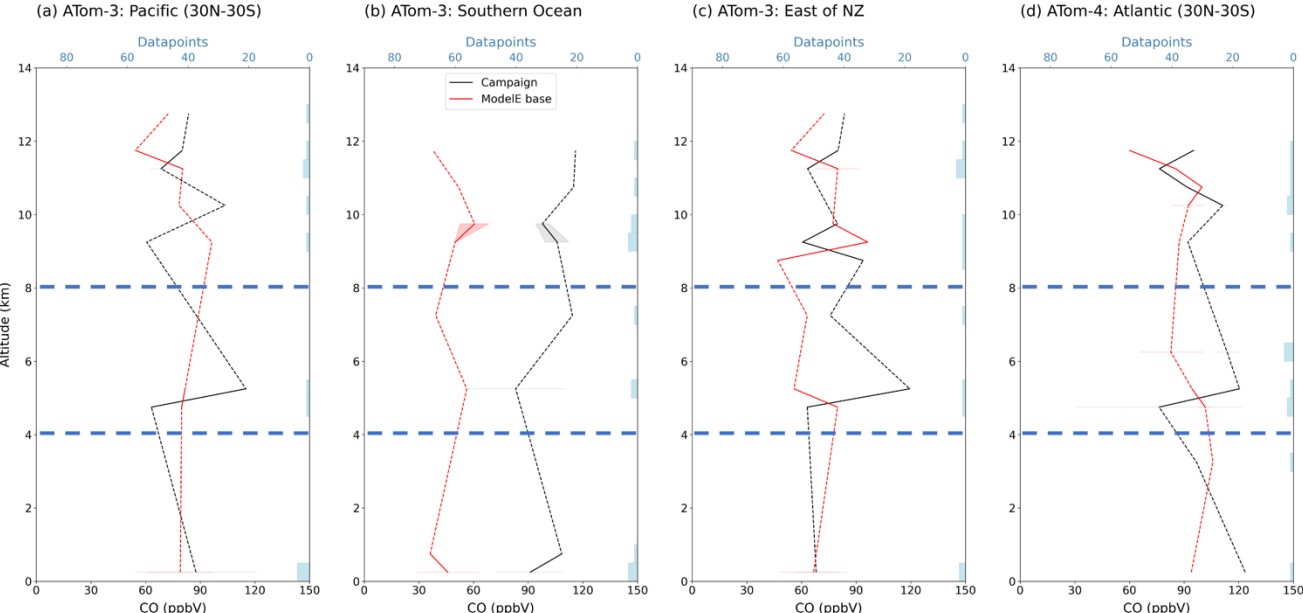

**Figure A2.** Vertical profiles of carbon monoxide (CO) concentration measured by flight campaigns (black) and simulated by ModelE's base case of BrC representation (red), over remote regions, reported in units of ppbV. Consistent with Fig. A1, campaign and specific region of analysis is indicated on the top left of each plot, the midpoint of altitude bins is plotted, dashed lines indicate altitudes with no data, shaded areas show variability of data in each altitude bin, blue bars show the number of datapoints averaged in each bin, horizontal blue dashed-lines separate the lower-, mid-, and upper-troposphere, and data is filtered for BrC-Abs outliers. Data is not filtered for BB number fraction.





**Figure A3.** Vertical profiles of total organic aerosol (OA) concentration measured by flight campaigns (black) and simulated by ModelE's base case of BrC representation (red), over land/near sources, reported in units of µg m⁻³. Model data has been multiplied by a CO and BC scaling factor (Eqs. 1-2). Consistent with Fig. A1, campaign and specific region of analysis is indicated on the top left of each plot, the midpoint of altitude bins is plotted, dashed lines indicate altitudes with no data, shaded areas show variability of data in each altitude bin, blue bars show the number of datapoints averaged in each bin, horizontal blue dashed-lines separate the lower-, mid-, and upper-troposphere, data is filtered for BrC-Abs outliers and BB number fraction (when relevant).





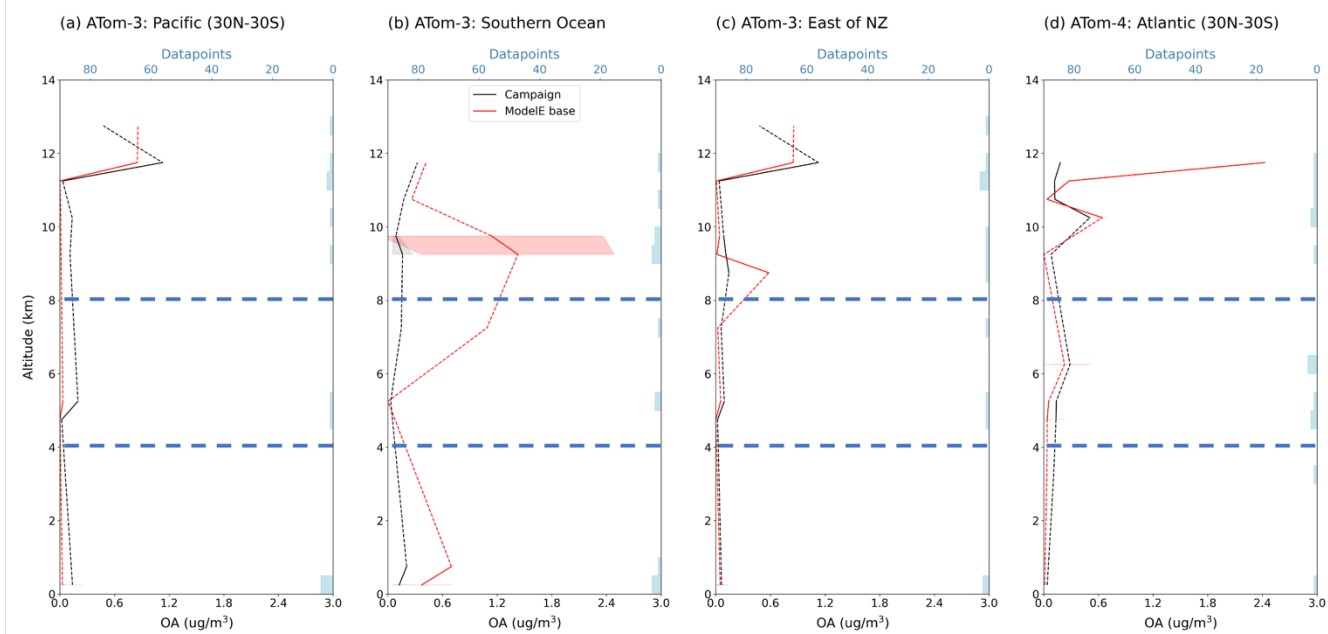

**Figure A4.** Vertical profiles of total organic aerosol (OA) concentration measured by flight campaigns (black) and simulated by ModelE's base case of BrC representation (red), over remote regions, reported in units of $\mu g\ m^{-3}$. Consistent with Fig. A3, a CO and BC scaling factor (Eqs. 1-2) have been applied to model data, campaign and specific region of analysis is indicated on the top left of each plot, the midpoint of altitude bins is plotted, dashed lines indicate altitudes with no data, shaded areas show variability of data in each altitude bin, blue bars show the number of datapoints averaged in each bin, horizontal blue dashed-lines separate the lower-, mid-, and upper-troposphere, and data is filtered for BrC-Abs outliers. It is not filtered for BB number fraction.





**Figure A5.** Vertical profiles of water-soluble organic carbon (WSOC) concentration measured by flight campaigns (black) and simulated by ModelE's base case of BrC representation (red), over land/near sources, reported in units of µgC m$^{-3}$. Consistent with Figs. A3, a CO and BC scaling factor (Eqs. 1-2) have been applied to model data, campaign and specific region of analysis is indicated on the top left of each plot, the midpoint of altitude bins is plotted, dashed lines indicate altitudes with no data, shaded areas show variability of data in each altitude bin, blue bars show the number of datapoints averaged in each bin, horizontal blue dashed-lines separate the lower-, mid-, and upper-troposphere, data is filtered for BrC-Abs outliers and BB number fraction (when relevant).



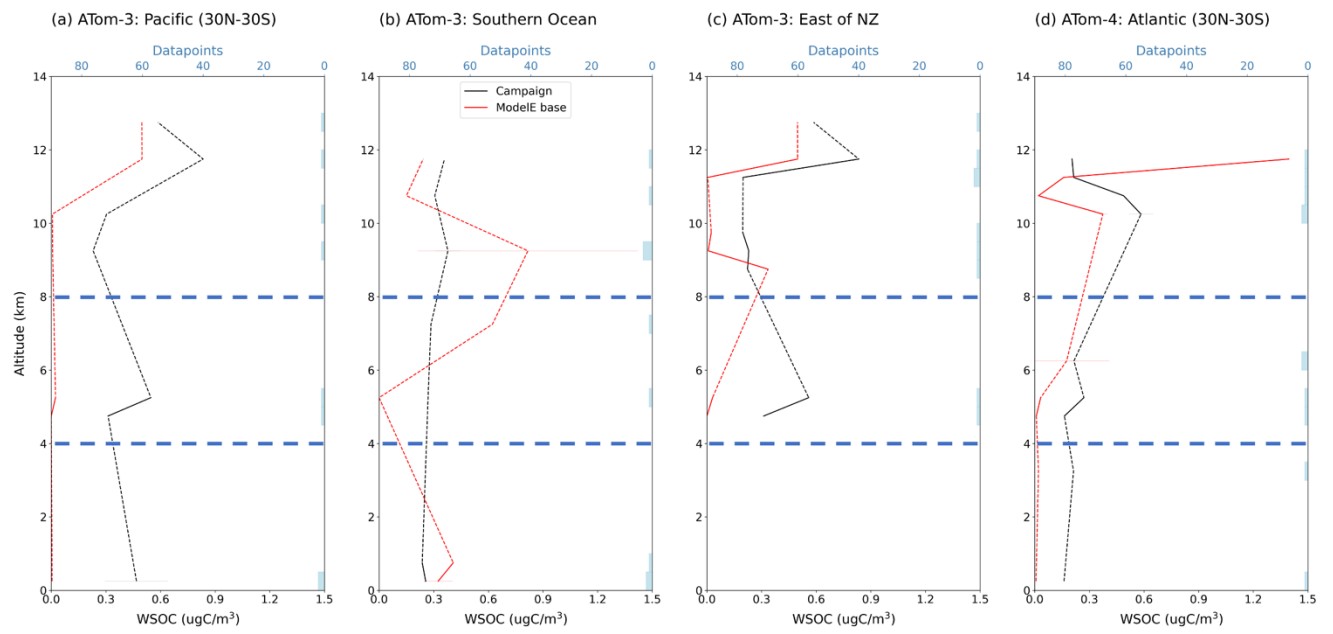

**Figure A6.** Vertical profiles of water-soluble organic carbon (WSOC) concentration measured by flight campaigns (black) and simulated by ModelE's base case of BrC representation (red), over remote regions, reported in units of µgC m$^{-3}$. Consistent with Figs. A4, a CO and BC scaling factor (Eqs. 1-2) have been applied to model data, campaign and specific region of analysis is indicated on the top left of each plot, the midpoint of altitude bins is plotted, dashed lines indicate altitudes with no data, shaded areas show variability of data in each altitude bin, blue bars show the number of datapoints averaged in each bin, horizontal blue dashed-lines separate the lower-, mid-, and upper-troposphere, and data is filtered for BrC-Abs outliers but not for BB number fraction.