# Peer review of "Exploring biases in brown carbon model representation with in-situ flight observations"

_EGUsphere, 2025_

## Referee Comment (RC1)

**Review of Delessio et al., "Exploring biases in brown carbon model representation with in-situ flight observations"**

There is considerable uncertainty in estimating the radiative forcing of brown carbon (BrC) in climate models, and assessing these uncertainties is crucial. Evaluating BrC modules using aircraft observations is particularly important (and currently underrepresented in existing studies), as these observations provide critical information on aged BrC in the upper atmosphere and remote regions—components that contribute significantly to the BrC's radiative effect. This manuscript aims to address the issue from this perspective, which carries certain scientific merit.

However, after carefully reading the current manuscript, one disappointing aspect is that, compared to the authors' previously published work on the BrC module in the GISS ModelE (Delessio et al., 2024a), the model representation remains outdated in some parts, and shows no clear improvements. It is understandable that there are large discrepancies between the current BrC module and observations, but in order to align with aircraft observations, the authors have introduced substantial calibration to the model's aerosol concentrations. These calibrations involve significant uncertainties, rendering the final conclusions unreliable.

As such, the evaluations presented in this manuscript do not provide reliable insights or constructive suggestions for model development. The manuscript lacks sufficient new information to qualify as a publishable journal article and instead reads more like a supplementary validation of the authors' earlier work. Although the paper conducts a series of evaluations—examining the impacts of adjusted emissions, BrC-to-OA ratios, and wet removal—it ultimately does not offer a best estimation for global glyoxal. The methodology overlaps heavily with the previous paper, and the study fails to produce scientifically sound or conclusive findings.

The authors need to substantially improve the model based on recent research, studies using other GCMs, and advancements in other aerosol modules of ModelE to make this study an independent scientific work.

The detailed concern and issues are listed below:

Line 199. Aromatics SOA contributed significantly (maybe the most) aged secondary BrC because its bleaching lifetime is long (~12 hours). Please justify how you deal with this underestimation.

Line 205-206. In DeLessio et al., 2024a, BrC emissions are parameterized based on BC-to-OA emission ratio, according to lab experiments. In this work, BrC-to-OA emission proportion of 35% is prescribed, without showing a reliable reference. It is confusing why this modification was applied.

Lines 215-220. The settings of BrC module is essential in this work, so it cannot be simply referred to a reference. On the other hand, there are confusing modifications in Line 205-206, so the authors need to clarify how the BrC module was built in ModelE.
Lines 234-235. Are BrC BB emissions prescribed in GFAS? It conflict with previous text.

Lines 236-238. There is a strong diurnal cycle of fire, for most of time, it seldom burn at night. Several studies have derived model usable fire diurnal profile to scale, for example that constrained

by FIREX (Tang et al., 2022, https://doi.org/10.1029/2022JD036650). This potentially leads to a 50% underestimation over the day.

Lines 244-246. The writing style is relatively informal in many places of the manuscript. This sentence is an example.

Lines 252-256. CO is a strong fire tracer, but There is also a strong anthropogenic (like industry and transportation) source. Although high concentrations of CO can be used to identify smoke, the ratio of CO cannot be used to scale fire emissions, especially over the clean region.

Line 259. Please report the MAE (mass absorption ratio) at a specific wavelength of different types of BrC in this work and compare to previous lab and observation studies.

Lines 283-285. The 'outliers' in the aircraft observations are fresh plumes. Removing that means the evaluation is not for plumes, then what part of BrC was evaluated in this work? Why 2 times standard deviation is a threshold? The model simulation cannot well fit the fresh plume, so other studies sometimes used hourly mean to compare, or use median + 25-75[th] quantiles to evaluate, instead of mean and standard deviations.